# Effects of Focused Vibrations on Human Satellite Cells

**DOI:** 10.3390/ijms23116026

**Published:** 2022-05-27

**Authors:** Silvia Sancilio, Sara Nobilio, Antonio Giulio Ruggiero, Ester Sara Di Filippo, Gianmarco Stati, Stefania Fulle, Rosa Grazia Bellomo, Raoul Saggini, Roberta Di Pietro

**Affiliations:** 1Department of Medicine and Ageing Sciences, “G. d’Annunzio” University of Chieti-Pescara, Via dei Vestini 31, 66100 Chieti, Italy; s.sancilio@unich.it (S.S.); sara.nobilio@gmail.com (S.N.); roger_93@virgilio.it (A.G.R.); gianmarco.stati@unich.it (G.S.); 2Department of Neuroscience Imaging and Clinical Sciences, “G. d’Annunzio” University of Chieti-Pescara, Via dei Vestini 31, 66100 Chieti, Italy; es.difilippo@unich.it (E.S.D.F.); stefania.fulle@unich.it (S.F.); 3StemTeCh Group, CAST, “G. d’Annunzio” University of Chieti-Pescara, Via Luigi Polacchi 11, 66100 Chieti, Italy; 4Department of Biomolecular Sciences, “Carlo Bo” University, Via Aurelio Saffi 2, 61029 Urbino, Italy; rosa.bellomo@uniurb.it; 5Department of Medical and Oral Sciences and Biotechnologies, “G. d’Annunzio” University of Chieti-Pescara, Via dei Vestini 31, 66100 Chieti, Italy; raoul.saggini@unich.it

**Keywords:** focused mechanoacoustic vibration, human satellite cells, aging, sarcopenia, atrophy, rehabilitation

## Abstract

Skeletal muscle consists of long plurinucleate and contractile structures, able to regenerate and repair tissue damage by their resident stem cells: satellite cells (SCs). Reduced skeletal muscle regeneration and progressive atrophy are typical features of sarcopenia, which has important health care implications for humans. Sarcopenia treatment is usually based on physical exercise and nutritional plans, possibly associated with rehabilitation programs, such as vibratory stimulation. Vibrations stimulate muscles and can increase postural stability, balance, and walking in aged and sarcopenic patients. However, the possible direct effect of vibration on SCs is still unclear. Here, we show the effects of focused vibrations administered at increasing time intervals on SCs, isolated from young and aged subjects and cultured in vitro. After stimulations, we found in both young and aged subjects a reduced percentage of apoptotic cells, increased cell size and percentage of aligned cells, mitotic events, and activated cells. We also found an increased number of cells only in young samples. Our results highlight for the first time the presence of direct effects of mechanical vibrations on human SCs. These effects seem to be age-dependent, consisting of a proliferative response of cells derived from young subjects vs. a differentiative response of cells from aged subjects.

## 1. Introduction

Skeletal muscle consists of long plurinucleate and contractile structures known as muscle fibers, originating from the fusion of myoblasts into plurinucleate myotubes during myogenesis [1]. Myofiber regeneration, during muscle repair, relies on its resident stem cells, also known as satellite cells, myogenic precursors of muscle tissue [2]. These adult stem cells are located between the basal lamina and the sarcolemma of the muscle fibers. SCs are typically quiescent in adult muscles. Following muscle injury, SCs are activated and stimulated to proliferate in the form of myoblasts, which migrate to the site of injury, and fuse with existing myofibers or differentiate to form new fibers [2,3]. A subpopulation of activated SCs, however, returns to quiescence to self-renew the SCs pool [4].

Aging is a complex process that, in multicellular organisms, results from the interplay among cells, intercellular communication, and systemic dysregulations, which coordinately compromise the homeostatic capacity of the organism. The aging of SCs is characterized by a decline in their number and functionality, due to a combination of factors, including defects in self-renewing mechanisms, loss of differentiation ability, apoptosis, and senescence [5]. Despite the huge number of studies, there is an intense debate in the scientific world to explain how age-related ineffective muscle regeneration is determined by changes in the extrinsic or intrinsic environments. These changes can inhibit the rege- nerative ability of otherwise competent SCs or render them less responsive to environmental cues, or rather, they can lead to a combination of both effects [6,7]. In our previous paper, we demonstrated the occurrence of spontaneous apoptosis in aged SCs cultured in vitro, supporting the hypothesis of intrinsic aging of human SCs [3]. The main consequence of SC aging is the reduced ability of muscle regenerative potential and progressive atrophy of skeletal muscle. This condition, known as sarcopenia, has important health care implications for humans, as it contributes to frailty, functional loss, and premature death [3]. Sarcopenia treatment is aimed at delaying its beginning or at least slowing down its progression. The most accredited therapies are based on physical exercise and nutritional plans, and they can be associated with rehabilitation programs, such as electrostimulation or vibratory treatment.

In rehabilitation, vibration has been used empirically since the 1960s, but only recently has the scientific community developed an interest in understanding its potential and mechanisms of action. In the therapeutic field, vibration is divided into whole-body vibration (WBV), applied to the whole body through oscillating platforms, and focal vibration (FV), in turn divided into mechanic FV (up to 100 Hz) and mechanoacoustic FV (up to 300 Hz), applied only to a specific muscle or muscle group. Thanks to the ability to direct the vibratory stimulus, FV can be used with high frequencies of around 300 Hz, avoiding at the same time the possible traumatic effects in surrounding tissues. Mechanoacoustic FV stimulates type III and IV mechanoreceptors, Pacini and Meissner’s corpuscles, and Golgi tendon organs [8] and can also activate the neuromuscular spindles afferent proprioceptive fibers, inducing tonic and involuntary muscle contractions, the so-called “vibratory tonic reflex” (VTR) [9]. This leads, in turn, to the activation of many α-motor neurons and the recruitment of previously inactive muscle fibers. The adaptive responses of neuromuscular apparatus to FV lead to increased contraction force, as in stimulated and adjacent muscles [10]. It is also known that mechanoacoustic FV acts on the endocrine system, where it seems to generate an increased production of testosterone and GH, associated with a reduction in cortisol [11]. Mechanotransduction can also start a consequential series of events leading to changes in transcription, transduction, and cell proliferation [12,13].

In a previous experimental rehabilitation protocol, we demonstrated that the 120 Hz mechanoacoustic FV application to subjects with myofascial problems promotes pain reduction and elasticity and improvement of tone and muscle stiffness [14]. Subsequently, we found that 300 Hz mechanoacoustic FV treatment leads to muscle tone normalization [15]. Furthermore, in aged subjects, mechanoacoustic FV allows the increase in muscle strength, postural stability, balance, and walking, leading to a reduction in the risk of falls and an increase in life quality [16,17]. However, in this scenario, the direct effects of mechanoacoustic FV on SCs are still unclear. Thus, this work was aimed at studying the changes induced by focused mechanoacoustic vibrations on young and aged SCs cultured in vitro to evaluate their effects on cell functions at multiple levels, involving proliferation and/or differentiation. In particular, we investigated the cell area and the presence of apoptotic, aligned, mitotic, and activated cells, comparing young and aged cells exposed to increasing vibratory treatment time intervals.

## 2. Results

### 2.1. ViSS (Vibration Sound System) Treatment Reduces the Percentage of Apoptotic Cells

The detection of DNA double-strand breaks with the terminal deoxynucleotidyl transferase dUTP nick-end labeling (TUNEL) technique evidenced positive nuclei inside both young and aged control SC-derived myoblasts. In particular, about 10–15% of apoptotic cells were found in young controls and more than 20% in the aged (Figure 1A).

Interestingly, whatever the ViSS treatment was, the percentage of apoptotic cells resulted to be greatly reduced (Figure 1A,B), with statistical significance when comparing control samples’ mean value with those treated for 20 min, whatever the age was (17.63 ± 5.94 control vs. 2.63 ± 1.48 20 min, *p* = 0.0469). Therefore, ViSS treatment seems to be related to a substantial reduction in the percentage of apoptotic cells in both young and aged samples, leading to a sort of protective effect. Furthermore, it is worth outlining that the labeling of actin cytoskeleton is more intense upon 10 min treatment in both age group samples in differentiation medium and upon 20 min treatment in samples cultured in proliferation medium.

### 2.2. ViSS Treatment Increases the Cellular Area, Number of Cells per Field, and Aligned Cells

As shown in Figure 2, aged SCs/myoblasts appeared generally featured with lower cell size and density compared to their young counterparts.

To assess any age- and treatment-related difference in cell size and number, we performed image analyses measuring cell area and counting the number of cells per field (10×) on digitally acquired images of hematoxylin–eosin-stained samples. By comparing young and aged myoblasts in control conditions, aged myoblasts showed a significantly reduced cell area in comparison with diff samples (Figure 3A). Already after 10 min, as well as after 20 and 30 min of ViSS treatment, there were no longer significant differences between young and aged groups. Interestingly, after ViSS treatment, both young and aged myoblasts showed a significantly increased cell size. In particular, both prol and diff aged cells almost doubled their cell area after 20 and 30 min of ViSS treatment (Figure 3B).

Regarding the number of cells per field, in control samples, there were no significant differences between young and aged SCs/myoblasts. After ViSS treatment, the number of cells per field increased in young samples compared with the aged samples, becoming significantly higher in young prol samples after 10 and 30 min treatment (Figure 4). Therefore, only in young SCs/myoblasts might ViSS treatment relate to a greater cellular density.

Moreover, we noticed a better cell alignment after ViSS treatment. Indeed, as shown in Figure 5A, we found an increased percentage of aligned cells in treated samples, with the highest percentage in aged diff samples after 20 min of ViSS treatment. In addition to the total quantification of aligned cells, these were classified into subgroups, based on the number of aligned cells identified in clusters. Six subgroups of clusters were recognized, starting with two cell clusters up to seven cell clusters. In Figure 5B, it can be noticed that aged samples show higher percentages of aligned cells organized in clusters of five, six, and seven cells. According to these results, ViSS could play an important role in favoring cell alignment and, thus, early-stage differentiation, especially in aged samples.

### 2.3. ViSS Treatment Increases the Mitotic Index

Given the greater cellular density (in the young) and the increased number of cells per field following treatments, we performed immunofluorescence analyses through specific single-molecule labeling (α-tubulin, pericentrin, and DAPI), which allowed us to calculate the mitotic index (MI) through the morphological identification, classification, and then quantification of mitotic events. As previously mentioned, each mitotic phase was determined through visual identification of specific morphological features identified with DAPI, α-tubulin, and pericentrin staining: condensed chromatin combined with double pericentrin spots in prophase, pericentrin asters migrated at cell poles with tubulin mitotic spindles in metaphase, chromatids pulled by microtubules towards cell poles in anaphase, loss of the spindle, formation of the nuclei, and visible cytokinesis in telophase (Figure 6A). As shown in Figure 6B, both young and aged samples display the presence of mitotic cells, mostly after 20 and 30 min of ViSS treatment, thus giving us a further chance to evaluate cellular proliferation.

Indeed, the MI mean value significantly increases after 20 and 30 min of ViSS treatment compared to control conditions in both prol and diff aged cells, starting with percentages of a maximum of 5% to reach the 20–25% after the longest time of treatment (16.83 ± 3.11 aged prol 20 min vs. 5.08 ± 4.53 control, *p* = 0.0236; 18.67 ± 3.22 aged prol 30 min, *p* = 0.0024) (Figure 7A).

Interestingly, early- and mid-phase mitotic cells (prophase, metaphase) were identified mostly after 20 min of ViSS treatment, while late-phase mitotic cells (anaphase, telophase) were especially found after 30 min of treatment (Figure 7B,C). Therefore, these results could indicate an important time-dependent proliferative effect of ViSS treatment on SCs/myoblasts.

### 2.4. ViSS Treatment Increases the Percentage of Activated Cells

Given SCs’ ability, recently found to express dystrophin only during their activation phase [18], the percentage of activated cells was evaluated through the visual detection of dystrophin-positive cells, which were identified through a spread fluorescent label with perinuclear cytoplasmic localization (Figure 8A).

As shown in Figure 8B, both young and aged groups display a significant increase in activated cells, mostly after 20 and 30 min of ViSS treatment in comparison with controls. Surprisingly, this effect is particularly evident in the aged group that starts with percentages of 8–9% prior to treatment to reach 15–16% after the longest time of treatment (Figure 8C). Consistently, a similar increase was found in aged samples labeled for Ki67, a well-known cell proliferation marker (Figure 9A). Interestingly, Ki67 labeling was found both in the nucleus and at the cytoplasmic level in both age groups of subjects and in both cell culture conditions (Figure 9B).

### 2.5. ViSS Treatment Influences Cell Motility and Myotube Formation

To analyze the process of myotube formation, a time-lapse microscopy video was performed on aged diff cells treated with ViSS for 30 min. The time-lapse video showed evident cell motility and, surprisingly, the formation of transient and unstable myotubes. The possible disassemble of a newly formed myotube is shown in the video frames reported in Figure 10: the marked cell primarily appears large and with a Y shape (Figure 10a); afterwards, the lower end of the cell shortens (Figure 10b–e) and then detaches from the rest of the cell, acquiring a smaller size and a roundish shape (Figure 10f–h).

## 3. Discussion

It is known that skeletal muscle mass, function, and repair capacity progressively decline with aging. It is widely accepted that SCs are necessary for muscle fiber regeneration, support repair, and remodeling, contributing to the maintenance of healthy muscle mass throughout life. Several studies have addressed the potential involvement of SCs in sarcopenia: their functional impairment and numerical reduction have been demonstrated because of age-associated extrinsic environmental changes, as well as cell-intrinsic changes [6,19]. In recent papers of our research group, we demonstrated the beneficial effects of FV on sarcopenic patients [14,15,16,17]. This kind of stimulation represents a safe, autonomous, and efficient way to increase or maintain muscle mass, strength, and function for aged and weak individuals, who are unable or unwilling to perform conventional workouts [20]. Furthermore, we also showed that neuromuscular electrical stimulation (NMES) improves skeletal muscle regeneration through SC fusion with myofibers in healthy aged subjects [21]. NMES is an effective passive protocol that can be used to induce local skeletal muscle contraction, due to its promotion of specific motor unit recruitment, through the administration of an electrical current [22].

Currently, most studies in the scientific literature have investigated the effects of ViSS mostly in vivo, but the possible role of vibration in improving muscle quality and function has not been fully elucidated, and, moreover, it remains unknown what changes vibration can promote on satellite cell activity. Therefore, this study was aimed at investigating the direct effects of FV on primary human SCs, isolated from young and aged human subjects and then cultured in vitro in both proliferation and differentiation medium.

One of the first important results that we found was the reduction in the number of apoptotic cells following ViSS treatment. In control conditions, aged samples showed higher percentages of apoptotic cells, in line with our previous work, demonstrating that aged SCs are more susceptible to apoptosis than their young counterparts [3]. It is inte-resting to note that the percentage of apoptotic cells in the aged SCs after ViSS treatment decreases to values equal or lower compared to their respective young counterparts (Figure 1A). These data support for the first time the hypothesis that ViSS treatment efficacy could also be due to an antiapoptotic protective mechanism, especially in aged subjects. To date, this assumption is quite different and innovative compared to what is known in the literature. Indeed, it has been reported that high-frequency mechanical vibration can successfully kill cancer cells but can also damage nearby normal healthy cells [23]. Furthermore, apoptosis, but not necrosis, has been found significantly increased at 48 h after mechanical vibration in human epidermoid carcinoma cell line A431 compared with cells maintained in static culture [24]. However, although the molecular pathways regulating caspase activity during apoptosis are well known, activated caspases do not always kill cells; in fact, they can also promote life instead of cell death in developmental and rege-nerative processes, as already demonstrated by our group in human SCs and by other authors in C2C12 cell lines [3,25]. Therefore, interestingly, the mechanical stimulus of vibration, possibly triggering the apoptosis pathway, could have led to nonapoptotic and regenerative effects on our cultured cells. Further investigations are needed to identify the molecular pathways involved in this response.

In parallel, the morphological analyses led us to further important results. In both young and aged treated samples, we found increased cell area (Figure 3) and percentages of aligned cells (Figure 5) and mitotic cells (Figure 7), with a greater significance in aged samples after 20 and 30 min of ViSS treatment. Due to the increase in the cellular area, it was quite common to observe larger irregular cells in our samples (Figure 2). However, even more often, the increased cell size underlies the elongation of the satellite cell during the development and alignment phase with other myoblasts, preparing the subsequent fusion phase for myotube generation. This result is also supported by the increase in the percentage of aligned cells (Figure 5A), which seem to be pushed by ViSS treatment in the subsequent differentiation phase. Regarding these features, it is well accepted that cytoskeletal rearrangements are essential for the proper regeneration of injured skeletal muscle. Several studies have shown the importance of microtubule dynamics for the maintenance and formation of skeletal muscle in human subjects and that remodeling of the cytoskeleton is pivotal for myoblast fusion and myotube formation [26]. Moreover, our data are in line with a recent study, where the effect of vibration (30 Hz) both in vivo and in vitro in animal models was investigated, showing that vibration promotes fusion of myoblasts, especially when applied directly to cultured cells [27].

Concerning SC/myoblast proliferation, we observed that the mitotic index rises in all experimental groups with the increase in treatment duration, in a time-dependent manner (Figure 7A), showing a statistically significant difference between controls and samples treated for 20 and 30 min, as well as between samples treated for 10 or 30 min. The increase in the mitotic index reaches its maximum peak after stimulation for 30 min, especially in aged cells in differentiation medium. Therefore, vibratory stimulation applied for 30 min at a frequency of 300 Hz could be the most effective protocol to induce SC/myoblast proliferation during in vivo therapy. Although in vitro cultured primary SCs detached from their physiological microenvironment, which may not completely reflect the complex muscle regeneration machinery occurring in living tissue, our results appear in line with the recent work of Usuki and colleagues [28] showing that the stimulation through mechanic waves can activate SCs, promoting their proliferation and facilitating in vivo muscle damage repair. In particular, an animal model was used to investigate the therapeutic effects of FV (90 Hz) on atrophic muscles due to immobilization. In this experimental model, the vibration, in addition to facilitating muscle trophism recovery, also restored the Pax7 downregulation induced by immobilization, thus increasing the number of SCs capable of being activated and proliferating.

Interestingly, we found in young samples, and not in the aged, an increase in the average number of cells per field (10×) after ViSS treatment (Figure 4). Consequently, the preponderant ViSS effect on young SCs/myoblasts would seem to be mainly in favor of proliferation. Although further studies are needed to investigate which molecular pathways might mediate this effect, this finding greatly correlates with clinical observations obtained in a young population (mean age: 15 years) affected with third-degree flat foot and hypotonia of the lower limb. Indeed, with a treatment protocol of high-frequency (300 Hz) ViSS FV (3 series of 30 min/week for 3 weeks), we found a 25% increase in the baseline value of the peak of maximum strength and a 22% increase in the total work ability of the lower limb recorded with isokinetic dynamometry (unpublished observations).

Concerning cellular activation, it is known that dystrophin is expressed in differen-tiated myofibers where it is required for sarcolemma integrity; however, it was recently found by Dumont and colleagues [4] that this molecule is also highly expressed in activated SCs, where it associates with the Ser/Thr kinase Mark2 (also known as Par1b), an important regulator of cell polarity. Indeed, they showed that the number of asymmetric divisions was strikingly reduced in dystrophin-deficient SCs, with the subsequent reduction of myogenic progenitors needed for proper muscle regeneration. Therefore, dystrophin has an essential role in the regulation of SC polarity and asymmetric division [18]. In our experimental model, we observed that the percentages of dystrophin-positive activated SCs in control conditions are slightly higher in aged samples. Then, the percentages were raised in all experimental groups with the increase in ViSS treatment duration, once again in a time-dependent manner and with stronger effects in aged samples (Figure 8). This result is consistent with data on cellular proliferation and discloses a possible role of ViSS treatment in promoting SC activation. This is particularly important if we consider that the level of dystrophin decreases with age. Indeed, an age-associated loss of dystrophin was recently observed in animal models and associated with an increase in membrane damage and neuromuscular junction instability [29]. Interestingly, this trend was confirmed by the labeling of Ki67, a known marker of cell proliferation, which increases in a time-dependent manner and particularly in aged samples committed to differentiation (Figure 9A). In fact, it is worth noting that this marker was located both at the nuclear level and, moreover, at the perinuclear and cytoplasmic levels in all the aligned cells forming myotubes (Figure 9B). These unprecedented observations of an extranuclear location of Ki67 in SCs are consistent with recent reports highlighting its role in cell cycle regulation and dramatic changes in its cellular distribution during cell cycle progression [30].

Finally, a very intriguing issue was raised by the time-lapse video performed on aged cells cultured in differentiation medium and treated with ViSS for 30 min (Figure 10). Indeed, we observed high cellular motility, together with the formation of unstable myotubes, which appear to form transiently and then disassemble. In this regard, we know that mammals possess a reduced ability to regenerate lost tissue compared with other vertebrates, which can regenerate through differentiation of precursor cells or de-differentiation. Mammalian multinucleated myotube formation is a differentiation process that arises from the fusion of mononucleated myoblasts, and it is thought to be an irreversible process toward muscle formation. However, it was found by Hjiantoniou and colleagues [31] that the overexpression of the Twist gene (a transcription factor known to inhibit differentiation of several cell types) in terminally differentiated myotubes induces reversal of cell differentiation. More specifically, this work provided strong evidence that the Twist gene can reverse differentiation, reinitiate the cell cycle, and cause morphological changes in myotubes, eventually leading them to cleave into smaller cell products, with also a reduction in the muscle transcription factor MyoD. Of note, Twist overexpression did not cause apoptosis in myotubes [31], supporting previous reports demonstrating that Twist might act to prevent apoptosis [32,33]. These findings are in line with our observations of reduced percentages of apoptotic SCs/myoblasts and increased cellular proliferation in vibrated samples highlighting the effects of ViSS on cell motility and migration capacity.

All in all, our in vitro results indicate for the first time the presence of direct effects of FV on human SCs. Interestingly, these effects seem to be age-dependent, assuming the mainly proliferative response of cells from young subjects vs. the mainly differentiative response of cells from aged subjects. This evidence adds new experimental knowledge to the already known neuromuscular effects in vivo [34] and is consistent with the age-specific protocols and therapeutic effects used and observed in rehabilitation, respectively. A wider cohort of subjects and further molecular investigations will be helpful to identify the mechanisms of action underlying these direct effects of vibration on human SCs.

## 4. Materials and Methods

### 4.1. Subjects and Satellite Cells Cultures

The SC populations were isolated from muscle fragments obtained from Vastus Lateralis biopsies of 3 healthy young subjects (23 ± 5 years old) and 3 healthy aged subjects (72 ± 9 years old) after written informed consent and approval from the Ethics Committee of the “G. d’Annunzio” University of Chieti-Pescara (Protocol Numbers: 1233/06 COET, dated July 25, 2006; 1634/8 COET, dated June 24, 2008; 1884 COET, dated May 15, 2009; 4/2016 COET, dated 25 February, 2016) [35]. The inclusion criteria were as follows: normal ECG and blood pressure; lack of bone and joint disorders, or metabolic (that is, diabetes) and/or cardiovascular diseases. The exclusion criteria were as follows: the presence of metabolic and/or cardiovascular diseases, evidence of hereditary or acquired muscular disorders, or psychiatric problems. No subject had been taking testosterone or other pharmacological therapies known to influence muscle mass. The first mononuclear cells migrated out of the explants within 7–13 days from the beginning of the culture (independent of donor age) and were removed. For cell growth, the myoblasts were cultured in growth medium (GM) constituted by HAM’s F-10 (Euroclone, Pero, MI, Italy) supplemented with 20% fetal bovine serum (FBS; Hyclone, Euroclone), 1% glutamax (Euroclone), 1% penicillin–streptomycin (Euroclone), and 50 µg/mL gentamicin (Euroclone) [36]. At the first cell passage when explants were removed, all cell populations were taken as “one population doubling level” (PDL). Cell populations were trypsinized and replated after reaching 50% confluency. The number of population doubling levels (expressed as PDL) at every passage was calculated as log N/ln 2, where N is the number of cells at the time of passage divided by the number of cells initially attached after seeding (Nt/Ni). Cultures were termed “senescent” when they failed to display one mean doubling after three weeks of refeeding [37]. The culture life span was 7–10 PDL for both young and aged samples.

Myoblasts were seeded on glass coverslips, coated with E-C-L collagen IV-laminin (EMD Millipore Corp., Merck KGaA, Darmstadt, Germany), placed in every single well of 24-well plates at a density of 2500–3000 cells/cm^2^, and maintained for 1 day in GM. Before cell treatment, 1 day after plating, in half both young and aged samples, the medium was replaced with differentiation medium (DM) consisting of DMEM high glucose (Euroclone) supplemented with 5% horse serum (Euroclone), 50 µg/mL of gentamycin (Euroclone), 10 µg/mL of insulin (SIGMA Chemical Co., St. Louis, MO, USA), 100 µg/mL of apo-Transferrin (Sigma), 1% glutamate (Euroclone), and 1% penicillin–streptomycin (Euroclone).

The cultures were assessed for the myogenic index by scoring the number of desmin-positive (Desm+) cells. Desmin expression was determined using the D33 antibody (Dako, Santa Clara, CA, USA, dilution 1:50) and immunostaining with the biotin streptavidin complex method (Dako), as described by Fulle et al. (2005). We counted 1500–2000 nuclei in 25–30 randomly selected fields. The mean values of 78.04 ± 13.02% Desm+ myoblasts in young subjects and 63.33 ± 20.6% in aged subjects were considered as standards for the experiments (Figure 11).

### 4.2. Cell Treatments and Experimental Design

SCs were treated with focused mechanoacoustic vibration administered through Vibration Sound System^®^ (ViSS) (Viessman s.r.l., Rome, Italy). It consists of a 32,000-revolution turbine with a flow rate of 35 m^3^/hour able to generate airwaves with a pressure up to 250 mbar and of a flow modulator, which makes the air vibrate with a pressure up to 630 mbar and a frequency up to 980 Hz producing mechanoacoustic waves. Vibration treatment was performed with increasing time intervals (10, 20, and 30 min) at a constant intensity (100 mbar) and frequency (300 Hz). The following experimental conditions were set up: young and aged SCs/myoblasts, both in proliferation (“young prol”, “aged prol”) and differentiation medium (“young diff”, “aged diff”), treated with and without focused mechanoacoustic vibration at increasing time intervals, respectively, at 10, 20, and 30 min. Then, cells were incubated in standard conditions, at 37 °C in a humidified atmosphere of 5% CO_2_, for 72 h before performing in vitro assays.

### 4.3. Immunofluorescent Staining of DNA Strand Breaks (TUNEL)

To visualize possible DNA damage caused by vibration treatment on a per-cell basis, we performed an in situ TUNEL assay, which detects single or double DNA strand breaks using labeled nucleotides polymerized to free 3′-hydroxyl termini in a reaction catalyzed by TdT. For the TUNEL assay, cells were fixed in 4% paraformaldehyde for 30 min at room temperature and incubated in a permeabilizing solution (0.1% Triton X-100, 0.1% sodium citrate) for 2 min on ice. Deoxyribonucleic acid strand breaks were identified with an in situ cell death detection kit (Roche, Basilea, Switzerland) according to the manufacturer’s instructions. Samples were counterstained using VECTASHIELD^®^ Antifade Mounting Medium with DAPI (Vector Laboratories, Burlingame, CA, USA) and observed with a ZEISS Axioskop 40 light microscope (Carl Zeiss, Gottingen, Germany) equipped with a CoolSNAP video camera (Photometrics, Tucson, AZ, USA) supported by Metamorph (Molecular Devices, Downingtown, CA, USA) software for acquiring digital images. The extent of DNA fragmentation was quantified through direct visual counting of green-fluorescent-labeled nuclei at 10× and 20× magnification. Apoptotic cells were scored out of a total of 70–130 cells from three different experiments for each experimental group. Positive control samples consisted of cells treated with deoxyribonuclease I at 2–5 mg/mL for 15 min at room temperature.

### 4.4. Histochemical Analyses

The cells were fixed with cold 80% ethanol for 30 min at 4 °C, dehydrated in a series of graded increases in alcohol concentrations, routinely stained with hematoxylin–eosin staining solution, and mounted in Bio Mount (Bio-Optica, Milano, Italy). All the samples were observed under a ZEISS Axioskop 40 light microscope (Carl Zeiss), equipped with a CoolSNAP video camera (Photometrics) supported by Metamorph (Molecular Devices) software for acquiring digital images. For cell area measurements, 12–30 cells from three different experiments for each experimental group were quantified at 20× magnification using ImageJ (National Institutes of Health, Bethesda, MD, USA) software. The number of cells per field was analyzed by counting hematoxylin-stained nuclei from 2–3 randomly selected fields at 10× magnification from three independent experiments for each experimental group. Aligned cells were quantified through direct visual counting of elongated, aligned cells at 10× and 20× magnification scored out of a total of 120–240 cells from three different experiments for each experimental group.

### 4.5. Immunofluorescence Analysis

All the samples were stained with indirect immunofluorescence. After fixation with 4% paraformaldehyde and blocking at 37 °C with 10% normal goat serum/PBS for 30 min, samples were incubated with: anti-β-actin mouse monoclonal (Sigma-Aldrich, St. Louis, MO, USA) antibody diluted 1:100 in 0.5% Tween 20 in 2% BSA/PBS overnight at 4 °C in combination with TUNEL assay; anti-pericentrin mouse monoclonal (Abcam, Cambridge, UK) and anti-α-tubulin rabbit polyclonal (Abcam) antibodies, respectively, diluted 1:500 and 1:1000 in 0.5% Tween 20 in 2% BSA/PBS overnight at 4 °C; anti-dystrophin (Abcam) antibody diluted 1:100 in 0.5% Tween 20 in 2% BSA/PBS overnight at 4 °C; anti-Ki67 (Dako) antibody diluted 1:50 in 0.5% Tween 20 in 2% BSA/PBS overnight at 4 °C. After several washings with PBS, samples were incubated with either goat anti-mouse IgG TRITC, donkey anti-rabbit IgG FITC (Jackson Immuno Research, West Grove, PA, USA), or goat anti-rat IgG Alexa Fluor^TM^ 488 (Invitrogen, Carlsbad, CA, USA), respectively, diluted 1:25, 1:50 and 1:50 in 0.5% Tween 20 2% BSA/PBS for 45 min at 37 °C. Nuclei were counterstained using VECTASHIELD^®^ Antifade Mounting Medium with DAPI (Vector Laboratories, Burlingame, CA, USA). The observations were carried out with a ZEISS Axioskop 40 (Carl Zeiss) light microscope equipped with a Coolsnap VideoCamera. Metamorph (Molecular Devices) software was used for acquiring computerized images. Cell mitotic phase was determined through visual identification of specific morphological features of samples stained with DAPI, pericentrin, and tubulin stainings (Figure 7A): condensed chromatin combined with double pericentrin spots in prophase; pericentrin asters migrated at cell poles with tubulin mitotic spindle in metaphase; chromatids pulled by microtubules towards cell poles in anaphase; loss of the spindle, formation of the two nuclei and visible cytokinesis in telophase. Cellular proliferation was measured in terms of mitotic index (MI), as a percentage defined by counting the number of cells undergoing mitosis divided by the total number of cells observed in randomly selected fields (i.e., number of mitotic cells/total number of cells × 100). Mitotic cells were identified at 20× and 40× magnification scored out of a total of 40 cells from three independent experiments for each experimental group. Cellular activation was quantified through direct visual counting of dystrophin-positive fluorescent labeled cells at 10× and 20× magnification. Activated cells were scored out of a total of 39–148 cells from three different experiments for each experimental group.

### 4.6. Time-Lapse Microscopy

The CytoSmart LUX10X device (CytoSMART Technologies BV, Eindhoven, the Netherlands) was used to monitor cultured SCs. Bright-field images were acquired in an incubator every 15 min for 72 h. Cell tracking (“tracking view” mode) was obtained through a novel contrast-based segmentation algorithm provided by CytoSMART Technologies BV. Image analysis was used to distinguish cells from the background. Contrast-limited histogram equalization was applied to the Gaussian-filtered images to remove uneven background shading and improve overall contrast.

### 4.7. Statistical Analysis

Data obtained were analyzed with GraphPad Prism version 8.0.0 (GraphPad Software, San Diego, CA, USA) software for Windows. Results are presented as mean  ± standard error of the mean (SEM). Statistically significant differences were determined with one-way ANOVA or two-way ANOVA (for grouped data) analysis of variance, followed by Tukey’s multiple comparisons test. Statistical significance was accepted at *p*  <  0.05.

## Figures and Tables

**Figure 1 ijms-23-06026-f001:**
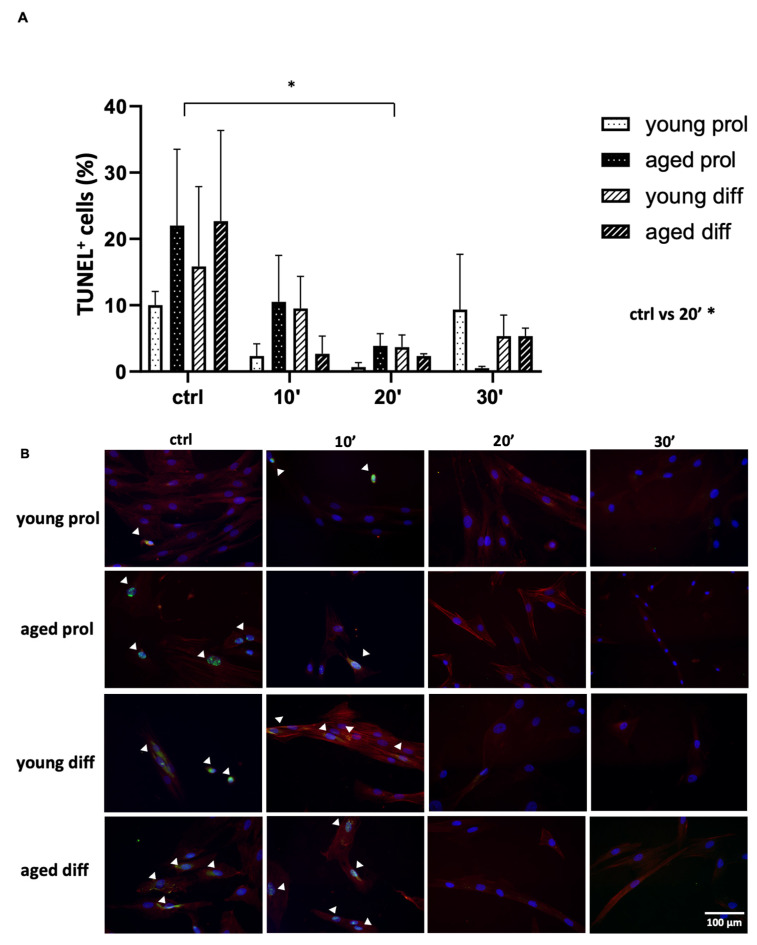
Effect of ViSS treatment on SC/myoblast apoptosis and actin cytoskeleton. (**A**) Data are expressed as mean ± SEM of three independent experiments (*n* = 3); apoptotic cells were scored out of a total of 70–130 cells for each experimental group; significance was determined with two-way ANOVA test, followed by Tukey’s multiple comparisons test (main row effect). * *p* ≤ 0.05. (**B**) Representative fields of TUNEL (green labeling) and β-actin (red labeling) doubled-stained SCs, counterstained with DAPI (blue labeling); white arrows point at apoptotic cells.

**Figure 2 ijms-23-06026-f002:**
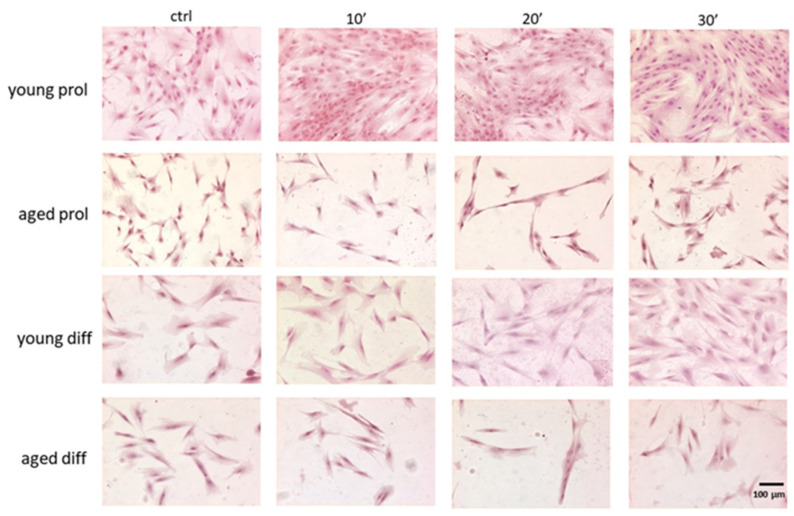
Effect of ViSS treatment on SC/myoblast size, density, and alignment. The cells typically showed an elongated morphology and the presence of one central nucleus and acidophilic cytoplasm. Representative fields of hematoxylin–eosin-stained samples.

**Figure 3 ijms-23-06026-f003:**
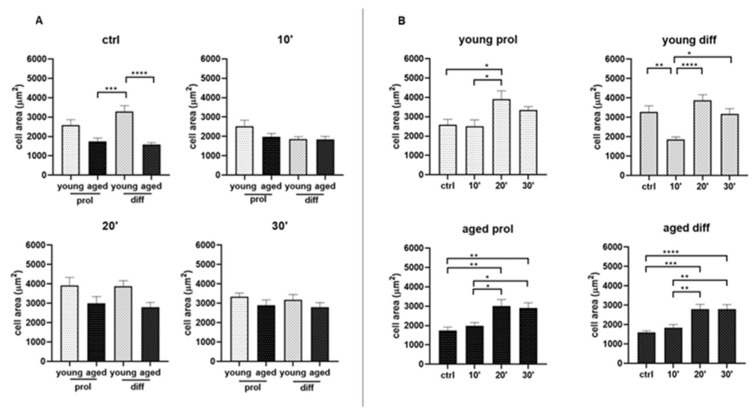
Effect of ViSS treatment on SC/myoblast area. (**A**) Comparison between young and aged samples in the same experimental conditions. (**B**) Control and treated samples of the same experimental group. Data from three different experiments are expressed as mean ± SEM. A total of 16–22 cells for each group per experiment were analyzed. Significance was determined with ordinary one-way ANOVA test, followed by Tukey’s multiple comparisons test. * *p* ≤ 0.05, ** *p* ≤ 0.005, *** *p* ≤ 0.001, **** *p* ≤ 0.0001.

**Figure 4 ijms-23-06026-f004:**
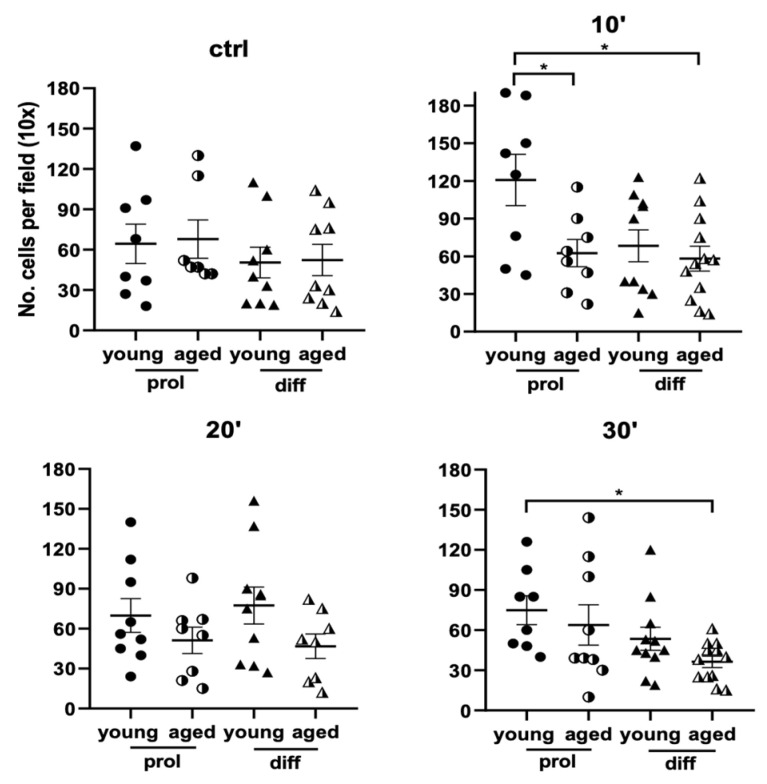
Effect of ViSS treatment on SC/myoblast number/field. Data from three different experiments are expressed as mean ± SEM. Hematoxylin-stained nuclei from 2 to 3 randomly selected fields for each group per experiment were counted at 10× magnification. Significance was determined with ordinary one-way ANOVA test, followed by Tukey’s multiple comparisons test. * *p* ≤ 0.05.

**Figure 5 ijms-23-06026-f005:**
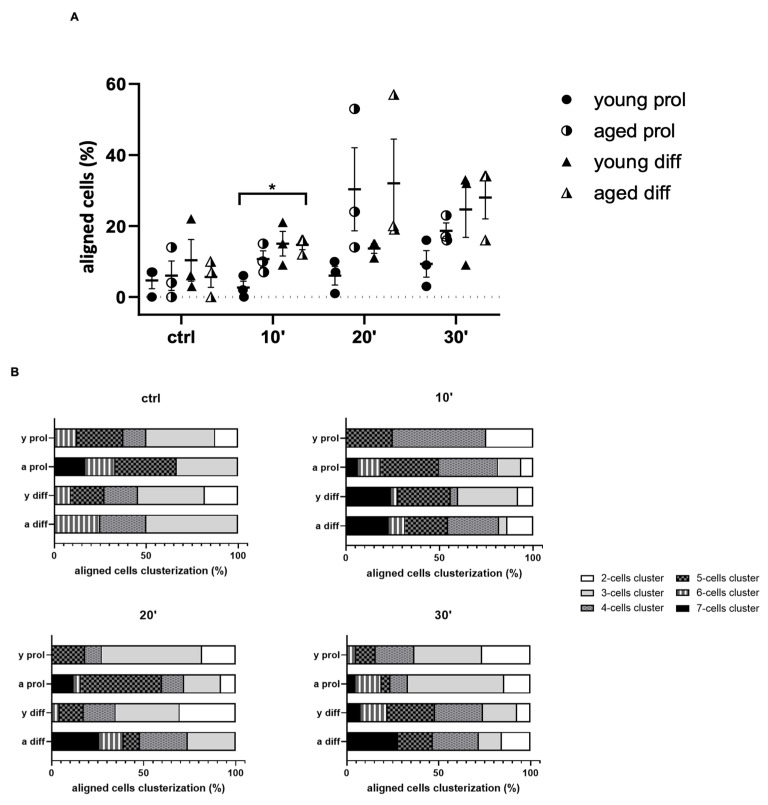
Effect of ViSS treatment on SC/myoblast alignment. (**A**) Percentages of aligned cells and (**B**) aligned cell clusterization. Data are expressed as mean ± SEM of three independent experiments (*n* = 3). Aligned cells were scored out of a total of 120–240 cells for each experimental group. Significance was determined with two-way ANOVA test, followed by Tukey’s multiple comparisons test (simple effects within rows). * *p* ≤ 0.05.

**Figure 6 ijms-23-06026-f006:**
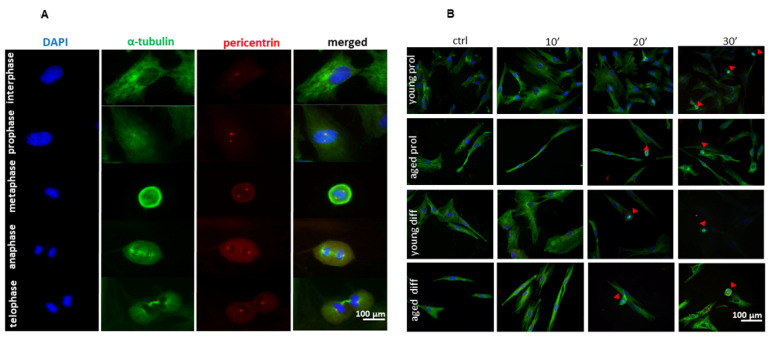
Distribution of mitotic phases in young and aged SCs/myoblasts after ViSS treatment. (**A**) Each mitotic phase was determined through visual identification of specific morphological features identified with DAPI, α-tubulin, and pericentrin labeling. (**B**) Representative fields from both age groups. Mitotic cells are marked with red arrows.

**Figure 7 ijms-23-06026-f007:**
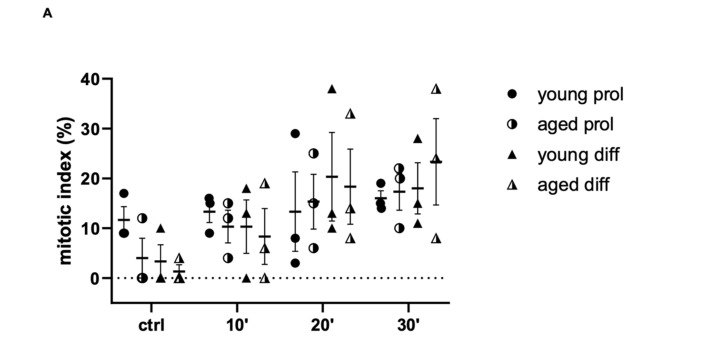
Effect of ViSS treatment on SC/myoblast proliferation. (**A**) Mitotic index (%). (**B**) Percentages of cells in the different mitotic phases. (**C**) Mitotic cell distribution in the different phases in young vs. aged samples. Data are expressed as mean ± SEM of three independent experiments (*n* = 3). Mitotic index was scored out of a total of 39–41 cells for each experimental group.

**Figure 8 ijms-23-06026-f008:**
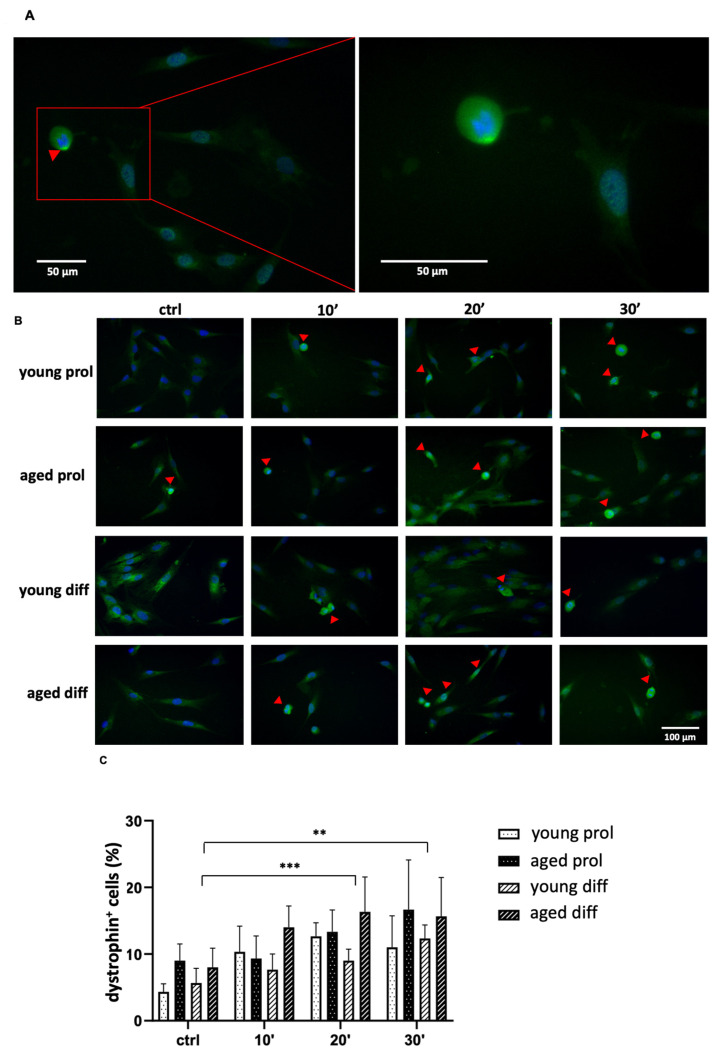
Effect of ViSS treatment on SC activation. (**A**) Green-fluorescent dystrophin-positive activated cell (red arrow). (**B**) Representative fields from both age groups, with red arrows pointing at dystrophin-positive activated cells. (**C**) Percentages of dystrophin-positive activated cells in both age groups. Data are expressed as mean ± SEM of three independent experiments (*n* = 3). Activated cells were scored out of a total of 39–148 cells for each experimental group. Significance was determined with two-way ANOVA test, followed by Tukey’s multiple comparison test (main row effect). ** *p* ≤ 0.005, *** *p* ≤ 0.001.

**Figure 9 ijms-23-06026-f009:**
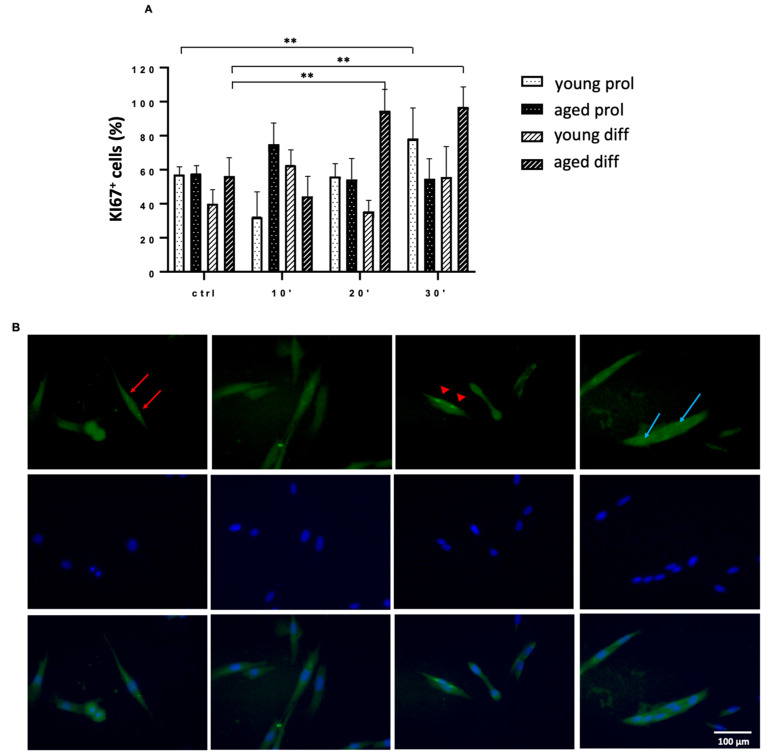
Effect of ViSS treatment on SC activation. (**A**) Data are expressed as percentage ± SEM of three independent experiments (*n* = 3) in comparison with age-matched controls. Ki67-positive cells were scored out of a total of 40–80 cells for each experimental group. Significance was determined with two-way ANOVA test, followed by Tukey’s multiple comparison test (main row effect). ** *p* ≤ 0.005. (**B**) Representative patterns of localization of green-fluorescent Ki67 labeling at perinuclear level (red arrows), nuclear level (red arrowheads), and at cytoplasm (pale blue arrows). In merge panels at the bottom, green fluorescence of Ki67 is overlapped with blue fluorescence of nuclear DAPI staining.

**Figure 10 ijms-23-06026-f010:**
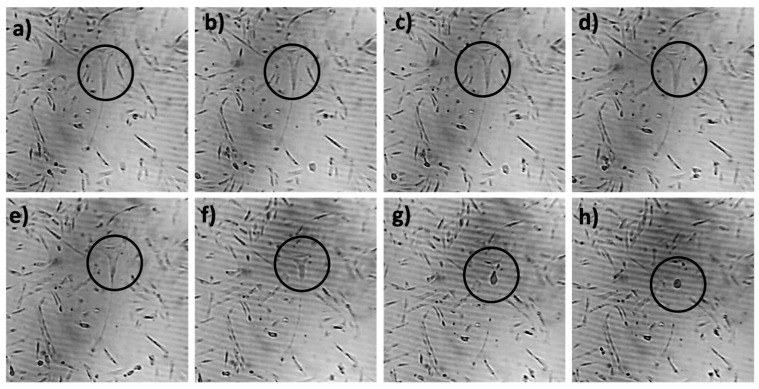
Sequential images of the time-lapse video (**a**–**h**). The possible formation of an unstable transient myotube is circled in black.

**Figure 11 ijms-23-06026-f011:**
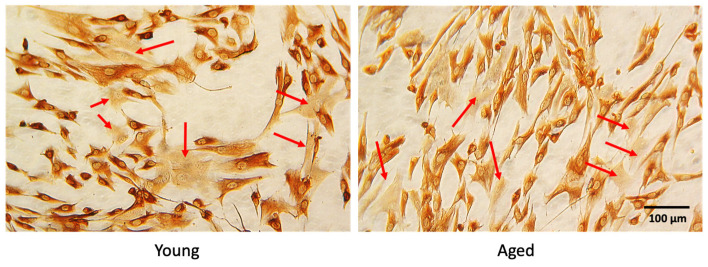
Representative images of desmin-positive (brown) and desmin-negative (arrows) cells obtained from young and aged subjects, viewed under an inverted microscope.

## Data Availability

Not applicable.

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
