# Peer review of "Effects of Focused Vibrations on Human Satellite Cells"

_ijms, 2022, doi:10.3390/ijms23116026_

Round 1

Reviewer 1 Report

Sancilio and collaborators investigated the direct effects of Focal Vibration (FV) on primary human SCs, isolated from young and aged human subjects and then cultured in vitro in both proliferation and differentiation medium. They performed cell stimulation in three intervals (10, 20, and 30 minutes) at constant intensity and frequency.

Interestingly, while many of the studies in the scientific literature have investigated the effects of ViSS especially in vivo, Sancilio and coworkers have tried to measure the changes focal vibration can promote directly on the activity of satellite cells.

After 72 hours they analyzed the effect of the treatment measuring many parameters: the percentage of apoptotic cells, the cell area, the mitotic index, the percentage of activated cells, the cell motility and the ability to form myotubes.

They found, in both young and aged SCs, reduced percentages of apoptotic cells, increased cell size and increased percentages of aligned cells, mitotic events, and activated cells.

The fact that in vitro results seem to indicate the presence of direct effects of FV on human SCs is noteworthy. Furthermore, these effects seem to be different in an age-dependent manner, with a greater proliferative effect in MPs from young donors and a greater differentiative response in MPs from aged donors

This work seems very interesting and the results presented are very convincing.In particular, the ability to induce a cellular response with a single treatment is a remarkable result. It requires further investigation to evaluate the pathway activated with this approach.

However, I think that the paper requires a minor review of some sections: in particular methods and results.

General comments:

-Introduction: well done and exhaustive but the phase that starts at row 50 and finishes at row 55 is too long and not clear.

-Results: generally, the experiments and the experimental methods are fully described but many images are not very clear and are difficult to read. I would suggest simplifying all the images that contain too much information and are too crowded. Furthermore in all images the authors name SCs while in the results they name the MPs

-Discussion: The discussion is broad and clearly written

-Bibliography: nothing to report

Specific comments

-fig1A the graph is too small and it is not clear which values are significant

-fig 4 I suggest arranging the graphs in two rows

-fig 5A the graph is too small

-fig 7A the graph is too small and it is not clear which values are significant

-fig 8 I suggest rearranging the three parts of this image to emphasize the graph

-in the results paragraph 2.2 the measured CSA values are missing

-row 414 I suggest to delete the phrase: (however, frequency within 300 Hz is recommended)

Author Response

Referee 1

Sancilio and collaborators investigated the direct effects of Focal Vibration (FV) on primary human SCs, isolated from young and aged human subjects and then cultured in vitro in both proliferation and differentiation medium. They performed cell stimulation in three intervals (10, 20, and 30 minutes) at constant intensity and frequency.

Interestingly, while many of the studies in the scientific literature have investigated the effects of ViSS especially in vivo, Sancilio and coworkers have tried to measure the changes focal vibration can promote directly on the activity of satellite cells.

After 72 hours they analyzed the effect of the treatment measuring many parameters: the percentage of apoptotic cells, the cell area, the mitotic index, the percentage of activated cells, the cell motility and the ability to form myotubes.

They found, in both young and aged SCs, reduced percentages of apoptotic cells, increased cell size and increased percentages of aligned cells, mitotic events, and activated cells.

The fact that in vitro results seem to indicate the presence of direct effects of FV on human SCs is noteworthy. Furthermore, these effects seem to be different in an age-dependent manner, with a greater proliferative effect in MPs from young donors and a greater differentiative response in MPs from aged donors

This work seems very interesting and the results presented are very convincing. In particular, the ability to induce a cellular response with a single treatment is a remarkable result. It requires further investigation to evaluate the pathway activated with this approach.

However, I think that the paper requires a minor review of some sections: in particular methods and results.

General comments:

-Introduction: well done and exhaustive but the phase that starts at row 50 and finishes at row 55 is too long and not clear.

We thank the referee for his/her comments. We have amended the phrase.

-Results: generally, the experiments and the experimental methods are fully described but many images are not very clear and are difficult to read. I would suggest simplifying all the images that contain too much information and are too crowded. Furthermore in all images the authors name SCs while in the results they name the MPs

As requested, we have simplified all the images and changed labels when appropriate.

-Discussion: The discussion is broad and clearly written

We thank the referee for his/her appreciation.

-Bibliography: nothing to report

Specific comments

-fig1A the graph is too small and it is not clear which values are significant

-fig 4 I suggest arranging the graphs in two rows

-fig 5A the graph is too small

-fig 7A the graph is too small and it is not clear which values are significant

-fig 8 I suggest rearranging the three parts of this image to emphasize the graph

All the figures have been changed according to the referee’s suggestions.

-in the results paragraph 2.2 the measured CSA values are missing

We did not put in the manuscript text the measured values of cell area, number of cells/field or aligned cells to avoid confusion and a heavy reading. All the measured values are reported in the relevant figures (Figs. 3, 4, 5).

-row 414 I suggest to delete the phrase: (however, frequency within 300 Hz is recommended)

Done.

Reviewer 2 Report

Ref. NO.: ijms-1690091

Title: Effects of focused vibrations on human satellite cells

Overview and general recommendation:

The aim of this study is to identify the focal vibration effect on satellite cells as the authors’ previous studies showed that focal vibration could enhance muscle strength and improves the condition of sarcopenia. There are works of literature revealing these vibrations cause the passive muscle contraction which induces tonic vibration reflex, thereby increasing the recruitment of motor unit. However, its long-term effect is still unknown. Therefore, it is indeed an interesting topic to explore as the results may clarify the long-term effect of vibration on satellite cells.

The whole experiment was carried out using the primary cultured myogenic cells from Vastus Lateralis muscle collected from young and aged donors and the immunohistochemistry experiments, histochemical analyses, and time-lapse observation was performed.
There is serious concern about the experiments that have been carried out. The comments in detail are as follows,

1. Figure 1, the author had claimed the effectiveness of Viss treatment on reducing the apoptosis of SCs and on the myogenic progenitors. However, the immunostaining had been performed with TUNEL assay only but no specific marker identifying the SCs or myogenic progenitors (e.g. Pax7 or Myf5 which are the well-known cell marker for SC and myogenic progenitor respectively). In other words, a double staining combining Pax7 and TUNEL should be performed. In fact, the author had yielded the effect on SC through many other experiments in this paper but all these data/experiments did not perform the Pax7 staining or the evidence to show they are truly SCs.

Figure 2 which is an imaging assay of primary cultured with HE staining but no immunostaining (myosin heavy chain) was performed to show these are truly matured skeletal muscle tissue. HE staining alone could not prove these are skeletal muscle tissue.

For the proliferation assay, the author performed the immunostaining using anti-dystrophin. However, the staining pattern is incompatible to what is used to known as the whole cell body appearing to be stained. I would recommend the author use the method of BrdU assay which is simple, clean, and less controversial.

4. in order to evaluate the capability of cell fusion, the author established a novel method of Fusion index by counting the nuclei within the myosin-heavy chain positive myofibers. However, the author can simply use Myogenin and/MRF4 to stain the myogenic cells that are under the fusion and count the positive cell number, since these are the two well-known markers for it.

Author Response

Title: Effects of focused vibrations on human satellite cells 

Overview and general recommendation:

The aim of this study is to identify the focal vibration effect on satellite cells as the authors’ previous studies showed that focal vibration could enhance muscle strength and improves the condition of sarcopenia. There are works of literature revealing these vibrations cause the passive muscle contraction which induces tonic vibration reflex, thereby increasing the recruitment of motor unit. However, its long-term effect is still unknown. Therefore, it is indeed an interesting topic to explore as the results may clarify the long-term effect of vibration on satellite cells.

The whole experiment was carried out using the primary cultured myogenic cells from Vastus Lateralis muscle collected from young and aged donors and the immunohistochemistry experiments, histochemical analyses, and time-lapse observation was performed.
There is serious concern about the experiments that have been carried out. The comments in detail are as follows,

1. Figure 1, the author had claimed the effectiveness of Viss treatment on reducing the apoptosis of SCs and on the myogenic progenitors. However, the immunostaining had been performed with TUNEL assay only but no specific marker identifying the SCs or myogenic progenitors (e.g. Pax7 or Myf5 which are the well-known cell marker for SC and myogenic progenitor respectively). In other words, a double staining combining Pax7 and TUNEL should be performed. In fact, the author had yielded the effect on SC through many other experiments in this paper but all these data/experiments did not perform the Pax7 staining or the evidence to show they are truly SCs.

We thank the referee for his/her comments and observations that allow us to better clarify the concept of SCs and myogenic progenitors. Indeed, SCs are Pax7-positive quiescent stem cells located between the sarcolemma and the basal lamina. During in vitro culture SCs become activated, upregulate MyoD, enter cell cycle and are defined myogenic progenitors including both cells in proliferation and cells forming myotubes, which, in turn, downregulate MyoD and upregulate Myog, regulatory factor of terminal differentiation. Thus, if the focus of our investigation had been to identify which cells were sensitive to the action of ViSS, we would have followed the expression of all these markers at the different time intervals in both age-groups. But, since the aim of the investigation was to look for any effects mediated by ViSS on SCs populations in the elderly, as nothing exists in literature, this type of investigation could be considered a second line study to deepen our insight on cells responsive to ViSS treatment. In addition, whenever SCs are put into culture they are routinely analyzed in our lab for the expression of desmin (as reported in the Materials and Methods section and in the new Fig. 11) and myogenic regulatory factors (Pax7, Myf5, MyoD, Myog) to validate the quality of the isolated population and follow-up their differentiation in culture. In the present investigation, we sticked to early time intervals (72 h) to look for early effects of ViSS treatment not only on apoptotic cell death but also on actin cytoskeleton, as commented in lines 143-146. Anyway, to meet the referee’s concerns, we have used the generic term SCs/myoblasts throughout the paper since we have not characterized the specific cell populations responsive to ViSS. To reassure the referee on the well-established method of SCs isolation and in vitro culture we are adding the list of already published papers of our research group:

  1. Fulle, S. Di Donna, C. Puglielli, et al., Age-dependent imbalance of the antioxidative system in human satellite cells, Exp. Gerontol. 40 (2005) 189e197.
  2. Trimarchi, A. Favaloro, S. Fulle, L. Magaudda, C. Puglielli, D. Di Mauro, Culture of human skeletal muscle myoblasts: timing appearance and localization of dystrophin-glycoprotein complex and vinculin-talin-integrin complex. Cells Tissues Organs 183 (2006) 87e98.

Lancioni H, Lucentini L, Palomba A, Fulle S, Micheli MR, Panara F. Muscle actin isoforms are differentially expressed in human satellite cells isolated from donors of different ages. Cell Biol Int. 2007 Feb;31(2):180-5. doi: 10.1016/j.cellbi.2006.10.002.

  1. Beccafico, C. Puglielli, T. Pietrangelo, R. Bellomo, G. Fanò, S. Fulle, Age-dependent effects on functional aspects in human satellite cells, Ann. N. Y. Acad. Sci. 1100 (2007) 345e352.
  2. Pietrangelo, C. Puglielli, R. Mancinelli, S. Beccafico, G. Fanò, S. Fulle, Molecular basis of the myogenic profile of aged human skeletal muscle SCs during differentiation, Exp. Gerontol. 44 (2009) 523e531.
  3. Beccafico, F. Riuzzi, C. Puglielli, et al., Human muscle satellite cells show age-related differential expression of S100B protein and RAGE, Age Dordr. 33 (2011) 523e541.

Di Filippo ES, Mancinelli R, Pietrangelo T, La Rovere RM, Quattrocelli M, Sampaolesi M, Fulle S. Myomir dysregulation and reactive oxygen species in aged human satellite cells. Biochem Biophys Res Commun. 2016 Apr 29;473(2):462-70. doi: 10.1016/j.bbrc.2016.03.030.

We hope to have clarified this issue.

Figure 2 which is an imaging assay of primary cultured with HE staining but no immunostaining (myosin heavy chain) was performed to show these are truly matured skeletal muscle tissue. HE staining alone could not prove these are skeletal muscle tissue.

As a matter of fact, we have done the immunolocalization of myosin heavy chain with MF20 monoclonal antibody but, as expected, we did not find a diffuse positivity of the cells due to the early time interval of observation (72 h) on which we focused our attention in the present study. In fact, at that time we could have detected only differences in gene expression but not in protein expression since 72 h are not enough for the cells to reach complete maturation. Nevertheless, it is worth outlining that aged diff samples display a higher positivity in comparison with young counterparts, in agreement with what we found with other techniques. On the other hand, morphological analysis with HE staining was able to display morphological features typical of human activated myoblasts: elongated morphology, presence of one or more central nuclei (in case of cell fusion), acidophilic cytoplasm. As detailed in the previous response, we have established since long the method of SCs isolation and primary culture, as the referee can see in the number of papers published by our research group. Anyway, if the referee considers it necessary, we can add to the paper the following Figure.

Representative fields of immunohistochemical detection of Myosin heavy chain with MF20 monoclonal antibody.

For the proliferation assay, the author performed the immunostaining using anti-dystrophin. However, the staining pattern is incompatible to what is used to known as the whole cell body appearing to be stained. I would recommend the author use the method of BrdU assay which is simple, clean, and less controversial.

If we understand well the referee’s comment, dystrophin staining is not easily interpretable. Thus, we enlarged all the 3 panels of Fig. 8 to better show which labelling was considered to score the cells as dystrophin-positive and thus activated cells as reported in literature (Dumont, N.A.; Wang, Y.X.; von Maltzahn, J.; Pasut, A.; Bentzinger, C.F.; Brun, C.E.; Rudnicki, M.A. Dystrophin expression in muscle stem cells regulates their polarity and asymmetric division. Nat Med 2015, 21, 1455-1463 doi: 10.1038/nm.3990). In fact, the absence of dystrophin has been related to the absence of asymmetric division of SCs which is necessary to lead to the formation of myogenic precursors. On the other hand, BrdU assay is certainly a consolidated method to identify cycling activated satellite cells but taking into consideration the 10 days allowed to submit a new version of the paper we could not carried out this assay implying at least 3 weeks for culturing and labelling cells. We, instead, detected in immunofluorescence the expression and localization of Ki67, well-known marker of cell proliferation. The results shown in the new Fig. 9 are consistent with dystrophin results.

  1. in order to evaluate the capability of cell fusion, the author established a novel method of Fusion index by counting the nuclei within the myosin-heavy chain positive myofibers. However, the author can simply use Myogenin and/MRF4 to stain the myogenic cells that are under the fusion and count the positive cell number, since these are the two well-known markers for it.

Concerning the fusion index, it is not a novel method, as the referee claims, since the method was published by our research group in 2005 (S. Fulle, S. Di Donna, C. Puglielli, et al., Age-dependent imbalance of the antioxidative system in human satellite cells, Exp. Gerontol. 40 (2005) 189e197) and related to the expression of Myogenin/MRF4 (F. Trimarchi, A. Favaloro, S. Fulle, L. Magaudda, C. Puglielli, D. Di Mauro, Culture of human skeletal muscle myoblasts: timing appearance and localization of dystrophin-glycoprotein complex and vinculin-talin-integrin complex. Cells Tissues Organs 183 (2006) 87e98). Anyway, we have deleted this part from the materials and methods section since this method is routinely used in our lab for SCs primary culture follow-up at 7 days which was not the case of the present investigation focused on an earlier endpoint (72 h).  

Round 2

Reviewer 2 Report

The author had answered all of my questions.